# Monitoring Patients with Metastatic Hormone-Sensitive and Metastatic Castration-Resistant Prostate Cancer: A Multidisciplinary Consensus Document

**DOI:** 10.3390/cancers11121908

**Published:** 2019-12-01

**Authors:** Alberto Lapini, Orazio Caffo, Giovanni Pappagallo, Roberto Iacovelli, Rolando Maria D’Angelillo, Vittorio Vavassori, Roberta Ceccarelli, Sergio Bracarda, Barbara Alicja Jereczek-Fossa, Luigi Da Pozzo, Giario Natale Conti

**Affiliations:** 1Department of Urology, University of Florence, University Hospital, 50134 Florence, Italy; lapinial@gmail.com; 2Department of Medical Oncology, Santa Chiara Hospital, 38112 Trento, Italy; 3Italian Society of Uro-Oncology, 40125 Bologna, Italy; giovanni.pappagallo@icloud.com (G.P.); roberta.ceccarelli@siuro.it (R.C.); 4Department of Medical Oncology, Fondazione Policlinico Universitario A. Gemelli IRCCS, 00168 Rome, Italy; roberto.iacovelli@policlinicogemelli.it; 5Radiotherapy Unit, Campus Bio-Medico University, 00128 Rome, Italy; R.Dangelillo@unicampus.it; 6Department of Radiation Oncology, Humanitas Gavazzeni, 24125 Bergamo, Italy; vittorio.vavassori@gavazzeni.it; 7Department of Medical Oncology, "Santa Maria" Hospital, 05100 Terni, Italy; s.bracarda@aospterni.it; 8Department of Oncology and Hemato-oncology, University of Milan, Division of Radiotherapy, IEO European Institute of Oncology, IRCCS, 20132 Milan, Italy; barbara.jereczek@ieo.it; 9Department of Urology, Università degli Studi Milano Bicocca - ASST Papa Giovanni XXIII, 24129 Bergamo, Italy; ldapozzo@hpg23.it; 10Urology Unit, Azienda Socio-Sanitaria Territoriale Lariana, Sant’Anna Hospital, 22042 Como, Italy; giario.conti@gmail.com

**Keywords:** castration-sensitive prostate cancer, castration-resistant prostate cancer, consensus conference, monitoring procedures

## Abstract

Background: The availability of a number of agents that are efficacious in patients with metastatic prostate cancer (mPC) has led to them being used sequentially, and this has prolonged patient survival. However, in order to maximize their efficacy, clinicians need to be able to obtain a reliable picture of disease evolution by means of monitoring procedures. Methods: As the intensive monitoring protocols used in pivotal trials cannot be adopted in everyday clinical practice and there is no agreement among the available guidelines, a multidisciplinary panel of Italian experts met to develop recommendations for monitoring mPC patients using a modified Delphi method. Results: The consensus project considered methods of clinically, radiographically, and biochemically monitoring patients with metastatic hormone-sensitive and metastatic castration-resistant prostate cancer undergoing chemotherapy and/or hormonal treatment. The panelists also considered the methods and timing of monitoring castration levels, bone health, and the metabolic syndrome during androgen deprivation therapy. Conclusions: The recommendations, which were drawn up by experts following a formal and validated consensus procedure, will help clinicians face the everyday challenges of monitoring metastatic prostate cancer patients.

## 1. Introduction

The prognosis of patients with metastatic prostate cancer (mPC) has dramatically improved over the last 10 years as a result of the introduction of a number of agents that are capable of significantly improving the overall survival of castration-resistant patients in everyday clinical practice. Patients with metastatic castration-resistant prostate cancer (mCRPC) can now be managed using two chemotherapeutic agents, docetaxel [1] and cabazitaxel [2]; two new androgen-receptor targeting agents (ARTAs), abiraterone acetate [3,4] and enzalutamide [5,6]; and one radiotherapeutic agent, radium 223 [7]. Improved survival has also been observed in patients with de novo metastatic castration-sensitive prostate cancer (mCSPC) treated with docetaxel [8,9].

In addition, the sequential use of these therapeutic options has further prolonged patient survival in comparison with both historical and current therapies [10,11,12], thus making it necessary to draw up specific monitoring protocols in order to permit the earlier identification of disease progression in patients with a lower tumor volume, whose better general condition allows them to receive successive therapeutic options.

The intensive monitoring protocols used in pivotal trials clearly cannot be adopted in everyday clinical practice, but there is still no agreement concerning the frequency or methods of monitoring CSPC and CRPC patients on and off treatment during the evolution of their disease treatment among the guidelines issued by the European Association of Urology (EAU), the American Urology Association (AUA), the European Society of Medical Oncology (ESMO), the Italian Association of Medical Oncology (AIOM), and the National Comprehensive Cancer Network (NCCN).

The situation is further complicated by the growing use of new imaging methods such as choline or PSMA positron emission tomography (PET) and whole-body magnetic resonance imaging (wbMRI). These seem to be more sensitive than traditional bone scintigraphy (BS) and computed tomography (CT), which are significantly limited in assessing disease burden particularly in the case of skeletal involvement. The new imaging methods could therefore improve our ability to detect and quantify the burden of bone and soft tissue metastases, but there is still no agreement as to how they can be used to evaluate and classify treatment responses, and the effective management of patients with advanced prostate cancer requires accurate, reproducible and validated methods.

In this complex clinical scenario, and with the participation of the Italian Association of Medical Oncology (AIOM), the Italian Association of Radiobiology (AIRB), the Italian Association for Radiation Oncology (AIRO), the Italian Society of Community Urologists (AURO.it), the Italian College of Chief Medical Oncologists (CIPOMO), and the Italian Society of Urology (SIU), the Italian Society of Uro-Oncology (SIUrO) organized a multidisciplinary expert consensus project with the aims of reviewing the available guidelines and evidence-based data, and making practical recommendations concerning the monitoring of patients with mCSPC and mCRPC.

## 2. Results

In this section the results are described via the statements discussed by the consensus panelists, which were summarized in the Table 1.

### 2.1. When Should Clinical and Biochemical Assessments Be Scheduled in the Case of an mCSPC Patient Who Is a Candidate for Androgen Deprivation Therapy (ADT) Alone?

With a consensus of 86%, the panelists recommended that:

The standard monitoring plan for an mCSPC patient who is a candidate for ADT alone should include a clinical ± biochemical assessment every 12 weeks for the first 12 months, and every 24 weeks thereafter.

### 2.2. When Should Imaging Assessments Be Scheduled in the Case of an mCSPC Patient Who Is a Candidate for ADT Alone?

With a consensus of 84%, the panelists recommended that:

Imaging assessments (preferably CT and BS) of an mCSPC patients who is a candidate for ADT alone should only be made in the case of a biochemical and/or clinical relapse.

### 2.3. Are There Any Factors that Could Influence the Baseline Monitoring Plan of an mCSPC Patient Who Is a Candidate for ADT Alone?

With a consensus of 90%, the panelists agreed that:

The factors that can individually change the initial monitoring plan of an mCSPC patient who is a candidate for ADT alone are age at the time of diagnosis, Gleason score, symptoms, the number and site(s) of metastases, the time of onset of metastases (at the time of diagnosis or during the course of progressive disease following radical local treatment), and the time interval between radical local treatment and the onset of metastases.

### 2.4. Are There Any Factors that Could Change the Initially Defined Monitoring Schedule of an mCSPC Patient Being Treated with ADT Alone?

With a consensus of 90%, the panelists agreed that:

The factors that may modify the monitoring schedule of an mCSPC patient being treated with ADT alone are the trend of PSA levels and disease-related symptoms, such as a worsening in performance status, the occurrence of a skeletal event, or a change in analgesic treatment.

### 2.5. When Should Clinical and Biochemical Assessments Be Scheduled in the Case of of an mCSPC Patient Who Is a Candidate for Treatment with ADT + Docetaxel?

Published evidence concerning the monitoring of such patients has been provided by the CHAARTED and LATITUDE trials [8,13]. CHAARTED randomized mCSPC patients to receive the combination of ADT + six courses of docetaxel or ADT alone: clinical evaluations were made at each treatment cycle (i.e., every 21 days), and biochemical evaluations were made at baseline and at cycles 2–6 [8]. LATITUDE randomized patients to receive ADT + abiraterone or ADT alone, and clinical and PSA assessments were made monthly [13].

With a consensus of 93%, the panelists therefore recommended that:

An mCSPC patient who is a candidate for treatment with ADT + docetaxel should be clinically evaluated at of each treatment cycle (i.e., every 21 days), whereas PSA levels should be repeated after at least the third and sixth treatment cycle.

### 2.6. When Should Imaging Assessments Be Scheduled in the Case of an mCSPC Patient Who Is a Candidate for Treatment with ADT + Docetaxel?

The CHAARTED trial did not provide any indications concerning imaging assessments during or after chemotherapy [8], but the LATITUDE trial included regular radiological assessments every four months [13]. In the case of abiraterone treatment, the definition of radiographic progression implies a knowledge of the best imaging response of mCSPC patients.

With a consensus of 91%, the panelists therefore recommended that:

An imaging assessment of a patient with mCSPC who is a candidate for treatment with docetaxel and ADT should be made at the end of docetaxel treatment using the same methods as those used at the time of the initial evaluation (preferably CT and BS).

### 2.7. Are There Any Factors that Could Influence the Baseline Monitoring Plan of an mCSPC Patient Who Is a Candidate for Treatment with ADT + Docetaxel?

There is very little evidence indicating whether there are factors that can be assessed before the start of treatment that could change previously planned biochemical, radiographic and clinical monitoring procedures. Disease volume is a well-established prognostic factor, as is the definition of high risk used in the LATITUDE trial, which was based on the number of bone metastases, Gleason score, and the presence of visceral metastases [13]. A retrospective analysis of 440 mCSPC patients receiving ADT alone has indicated advanced age, pain, a Gleason score of ≥8, and nadir PSA levels after ADT as prognostic factors [14]. It is worth noting that a decision to start docetaxel should not be based on changes in PSA levels during ADT because there are no prospective data indicating that this strategy is beneficial for patients.

With a consensus of 81%, the panelists therefore agreed that:

That there is no factor that should modify the standard monitoring schedule of an mCSPC patient who is a candidate for treatment with ADT + docetaxel.

### 2.8. Are There Any Factors that Could Change the Initially Defined Monitoring Schedule of an mCSPC Patient Being Treated with ADT + Docetaxel?

Published evidence, which mainly relates to mCRPC patients, indicates that an increase in PSA levels alone is not sufficient to indicate disease progression [15] as a significant increase can be observed in about 14% of patients during the first cycles of docetaxel treatment [16]. Patient assessments should always include an evaluation of pain because increased pain is frequently associated with disease progression.

With a consensus of 85%, the panelists therefore agreed that:

Increasing PSA levels and worsening disease-related symptoms (e.g., worsening performance status, the occurrence of a skeletal event, increased analgesic treatment) may require an earlier re-evaluation of an mCSPC patient being treated with ADT + docetaxel than that laid down in the initial monitoring plan.

### 2.9. When Should Clinical and Biochemical Assessments Be Scheduled in the Case of an mCSPC Patient without Progressive Disease Who Has Concluded Docetaxel Treatment but Is Continuing ADT?

The rate of progression in the STAMPEDE study, which tested the benefit of adding docetaxel to ADT alone in patients with mCSPC, was about 25% after 12 months of treatment and 40% after 24 months of treatment [17]. After completing chemotherapy, the patients in the CHAARTED trial underwent a clinical evaluation and PSA assessment every 12 weeks, whereas no regular imaging assessment was planned [8]; a clinical examination and changes in PSA levels were therefore considered sufficient to define progressive disease.

With a consensus of 92%, the panelists consequently recommended that:

The standard monitoring plan of an mCSPC patient without progressive disease who has concluded docetaxel treatment but is continuing ADT should include clinical and biochemical assessments at least every 12 weeks.

### 2.10. When Should Imaging Assessments Be Scheduled in the Case of an mCSPC Patient Who Has Concluded Docetaxel Treatment but Is Continuing ADT?

In accordance with the strategy adopted in the CHAARTED trial and with a consensus of 89%, the panelists recommended that:

A radiographic assessment (preferably CT and BS) of an mCSPC patient who has concluded docetaxel treatment but is continuing ADT is only required in the case of clinical and/or biochemical progression.

### 2.11. Are There Any Factors that Could Influence the Baseline Monitoring Plan of an mCSPC Patient Who Has Concluded Docetaxel Treatment but Is Continuing ADT in the Absence of Progressive Disease?

On the basis of experience and with a consensus of 92%, the panelists agreed that:

The factors that may modify the monitoring schedule of an mCSPC patient who has concluded docetaxel but is continuing ADT are PSA level, the appearance of symptoms, and the biological/clinical aggressiveness of the disease.

### 2.12. Are There Any Factors that Could Change the Initially Defined Monitoring Schedule of an mCSPC Patient Undergoing ADT Who Has Been Previously Treated with Docetaxel?

The occurrence of events such as an increase in PSA levels and/or the onset of pain in an mCSPC patient undergoing ADT who has been previously treated with docetaxel should obviously lead to the timing of the initially scheduled assessments being brought forward. At the same time, the presence of features indicating particularly aggressive disease, such as low PSA values in the presence of a tumour with a high Gleason score, should suggest regular radiographic as well as clinical and biochemical assessments.

With a consensus of 85%, the panelists accordingly agreed that:

The factors that may modify the monitoring schedule of an mCSPC patient undergoing ADT who has been previously treated with docetaxel are an increase in PSA levels and/or the onset or worsening of disease-related symptoms such as a worsening performance status, the occurrence of a skeletal event, an increase in pain therapy.

### 2.13. When Should Clinical Assessments Be Scheduled in the Case of an mCRPC Patient Who Is a Candidate for Chemotherapy?

On the basis of experience and with a consensus of 85%, the panelists recommended that:

The standard monitoring plan of an mCRPC patient who is a candidate for chemotherapy should include a clinical assessment at every cycle.

### 2.14. When Should Biochemical Assessments Be Scheduled in the Case of an mCRPC Patient Who Is a Candidate for Chemotherapy?

On the basis of experience and with a consensus of 85%, the panelists recommended that:

The standard monitoring plan of an mCRPC patient who is a candidate for chemotherapy should include a PSA assessment at least every 6–8 weeks.

### 2.15. When Should Imaging Assessments Be Scheduled in the Case of an mCRPC Patient Who Is a Candidate for Chemotherapy?

On the basis of experience and with a consensus of 85%, the panelists recommended that:

The first imaging assessment of an mCRPC patient who is a candidate for chemotherapy should be made after about 12 weeks using the same methods as those used for the baseline assessment (preferably CT and BS).

### 2.16. Are There Any Factors that Could Change the Initially Defined Monitoring Schedule of an mCRPC Patient during Docetaxel Treatment?

On the basis of experience and with a consensus of 93%, the panelists agreed that:

The factors that can modify the monitoring schedule of a patient with mCRPC receiving docetaxel treatment are an increase in PSA levels and the onset or worsening of disease-related symptoms such as a worsening performance status, the occurrence of a skeletal event, and an increase in pain therapy.

### 2.17. When Should Imaging Assessments Be Scheduled in the Case of an mCRPC Patient Who Has Completed Chemotherapy and Shows No Signs of Progression?

On the basis of experience and with a consensus of 84%, the panelists recommended that:

Imaging assessments of an mCRPC patient who has completed chemotherapy and shows no signs of progression should not be pre-planned, but depend on the results of clinical/biochemical assessments; in any case, it is recommended to use the same methods as those used for the baseline assessment (preferably CT and BS).

### 2.18. When Should Clinical and Biochemical Assessments Be Scheduled in the Case of an mCRPC Patient Who Is a Candidate for ARTA Treatment?

ARTA-treated patients with mCRPC are usually clinically evaluated every four weeks in order to monitor toxicity and assess the onset of treatment-related adverse symptoms. In relation to biochemical monitoring, the two pivotal studies of ARTAs in this setting [6,18] planned an initial assessment three months after starting ARTA treatment, and subsequent assessments every 4–8 weeks during the first year and every 12 weeks thereafter.

With a consensus of 89%, the panelists accordingly recommended that:

The standard follow-up schedule of an mCRPC patient who is candidate for ARTA should include a PSA assessment every 12 weeks and a clinical evaluation every four weeks.

### 2.19. When Should Imaging Assessments Be Scheduled in the Case of an mCRPC Patient Who Is a Candidate for Treatment with an ARTA?

The two pivotal trials [6,18] planned CT and BS assessments at baseline, and subsequently every two months for the first year, and every three months thereafter. As the median duration of radiological progression-free survival recorded in these trials ranged from 16 to 20 months, it would be advisable to make a first reference imaging assessment within the first 12 months of ARTA treatment in order to be able to evaluate any subsequent disease progression. The need for imaging assessments should be determined on the basis of the findings of clinical/biochemical assessments, and carried out using the same methods as those used for the baseline assessment (preferably CT and BS). Biochemical and clinical progression usually precede radiological progression during ARTA treatment (median onset about 12 vs. 16–20 months), although radiographic progression in the absence of biochemical progression was recorded in 25% of the patients participating in the PREVAIL study [19]. There was below-threshold consensus (77%) for the inclusion of a predefined instrumental follow-up schedule (twice a year), but a consensus of about 90% for the recommendation that:

The need for imaging assessments should be based on the findings of clinical/biochemical assessments.

### 2.20. Are There Any Factors that Could Influence the Baseline Monitoring Plan of an mCRPC Patient Who Is a Candidate for ARTA Treatment?

Evidence of more aggressive disease can certainly change the initially planned monitoring schedule of an mCRPC patient undergoing systemic ARTA treatment. The benefits of ARTA in docetaxel-naïve mCRPC patients are generally greater than those recorded in patients who have previously received docetaxel [3,4,5,20]. Data from the COU-AA 301 study [21] highlight the prognostic value of performance status (ECOG 2 vs. 0–1), serum albumin levels, and liver and bone metastases. There was below-threshold consensus (only 65%) for considering the site of metastases together with the treatment line (pre- vs post-docetaxel), but 85% consensus for the recommendation that:

The site of metastases and disease-related symptoms should be considered factors that may modify an initial monitoring schedule.

### 2.21. Are There Any Factors that Could Change the Initially Defined Monitoring Schedule of an mCRPC Patient Undergoing ARTA Treatment?

The frequency of assessments initially planned during systemic ARTA treatment may be changed in the case of suspected disease progression in order to stop potentially ineffective treatment. In this regard, the EAU guidelines highlight the importance of the presence of disease-related symptoms. In addition, the panelists at the 2017 St. Gallen Consensus Conference stressed that, regardless of its kinetics, rapid PSA progression combined with other factors may indicate a worse prognosis [22]. For these reasons, and with a consensus of 89%, the panelists recommended that:

When deciding on changes in the frequency of assessments of an ARTA-treated mCRPC patient, the trend of PSA levels and the onset of disease-related symptoms should be considered.

### 2.22. When Should Testosterone Assessments other than the Baseline Assessment Be Scheduled in the Case of Patients with Advanced Prostate Cancer (mCSPC/mCRPC)?

The aim of ADT is to maintain suppressed testosterone levels of <50 ng/dL (1.7 nmol/L). The initial phase of chemical castration is closely related to the reduction in PSA levels and so, as underlined by the EAU guidelines, testosterone suppression should be assessed in the case of biochemical progression. With a consensus of 89%, the panelists recommended that:

The standard monitoring plan of a patient with advanced prostate cancer undergoing ADT should include a testosterone evaluation every time there is an increase in PSA levels.

### 2.23. When Should Bone Health Assessments Other than the Baseline Assessment Be Scheduled in the Case of Patients with Advanced Prostate Cancer (mCSPC/mCRPC)?

All mPC patients undergo ADT, which may have adverse effects on bone health and the cardiovascular system. It is estimated that a patient undergoing ADT may have a lumbar mineral density loss of 4.6% per year, and that there is a possibility of developing a bone fracture in up to 14% of cases [23]. With a consensus of 89%, the panelists therefore recommended that:

The standard monitoring plan of a patient with advanced prostate cancer (mCSPC/mCRPC) should include regular bone health assessments.

### 2.24. When Should Assessments of Metabolic Alterations Other than the Baseline Assessment Be Scheduled in the Case of Patients with Advanced Prostate Cancer (mCSPC/mCRPC) Treated with ADT?

Hypogonadism secondary to ADT may lead to insulin resistance and the consequent onset of metabolic syndrome [24]. Furthermore, the prolonged use of ADT has been associated with the onset of thrombotic and ischemic (27%), as well as cardiovascular events (10%) [23]. The latter are certainly more likely in patients who have previously experienced a cardiovascular event [25].

With a consensus of 94%, the panelists therefore recommended that: 

Patients with advanced prostate cancer (mCSPC-mCRPC) treated with ADT should undergo regular metabolic assessments, particularly those at increased cardiovascular risk.

## 3. Discussion

Disease monitoring is one of the greatest challenges facing the clinicians who treat mPC patients because the possibility of sequentially administering the agents that have proved to be efficacious requires monitoring methods capable of providing a reliable picture of disease evolution. The ability to detect disease progression is crucial to enabling clinicians to stop an ineffective treatment that could lead to unnecessary side effects, and allowing them to propose a further treatment line that may be more efficacious in controlling the disease.

The concept of disease progression in patients with mCRPC has been clearly defined by the Prostate Cancer Working Group, which has modified its definition over time in order to keep up with advances in our knowledge of disease biology and the introduction of new therapeutic options [15]. The PCWG recommendations were mainly developed for clinical trials, but they are also valuable in everyday clinical practice.

The disease status of an mCRPC patient is defined on the basis of three factors: clinical status (based on the evaluation of disease- and treatment-related symptoms), PSA levels (the trends of which need to be cautiously interpreted), and radiographic changes (which require the definition of clear progression criteria).

Over the last 20 years, the assessment of PSA levels has been the mainstay of the management of prostate cancer patients, and biochemical responses or progression have driven the therapeutic choices of clinicians. It has already been established that changes in PSA levels reflect changes in the disease during the early stages of prostate cancer, but this is questionable in the castration-resistant phase. Previous editions of the PCWG guidelines have highlighted the fact that PSA flares may occur during the early courses of chemotherapy, and it is recommended that discontinuing treatment on the basis of increasing PSA levels alone should be avoided during the first 12 weeks [26]. More recently, the role of PSA during treatment with ARTAs has also been revised on the basis of evidence showing that radiographic progression can occur in the absence of biochemical progression [19].

As a result, clinical and imaging assessments have taken on a greater role in defining disease status. The latest version of the PCWG guidelines distinguishes the first evidence of progression from a clinical need to change or discontinue treatment by introducing the concept of clinical benefit; moreover, they also underline the importance of separating progression in existing lesions from the development of new lesions [15].

In this complex scenario, planning careful and regular monitoring is a crucial means of ensuring that patients receive active agents for as long as they really control the disease, and that clinicians view the monitoring of progression status in the light of adopting new therapeutic options.

Unfortunately, the PCWG guidelines do not provide any indications concerning the optimal timing of planned monitoring assessments, and the same is true of the prostate cancer guidelines issued by the main scientific societies. Furthermore, the monitoring plans used in the pivotal trials of agents active in mCSPC and mCRPC were designed to respond to trial and regulatory needs, and cannot be directly translated into everyday clinical practice. Finally, as the evidence provided in the literature is very limited and often confusing, clinician choices are frequently only based on their personal experience, which clearly does not guarantee optimal patient management as it may lead to the early discontinuation of a treatment in the absence of true progression, or the needless continuation of ineffective treatment because true progression is not recognized.

The use of a formal method of developing recommendations on the basis of the consensus of experts is therefore one of the best ways of addressing some aspects of a scenario devoid of evidence, such as that of monitoring mPC patients.

The experts’ recommendations concerning the evaluation of clinical status and PSA levels varied mainly on the basis of treatment: it is suggested that patients with mCSPC treated with ADT alone (at least in the first year of treatment) or receiving ADT after docetaxel treatment should undergo a clinical assessment every 12 weeks, whereas patients receiving docetaxel should be clinically evaluated at each treatment course and undergo less frequent biochemical assessments.

There were similar differences in the recommendations made for mCRPC patients treated with an ARTA or docetaxel. In the case of chemotherapy, the recommendations clearly reflect the everyday practice of clinically evaluating patients at each course in order to assess not only disease-related symptoms, but also treatment-related side effects. In the case of ARTA-based treatment, the suggestions underline the need to avoid the risk of monitoring the disease simply on the basis of serial PSA assessments, and indicate that, albeit less frequently, clinical assessments should be regularly planned in order to be able to capture signs of progressive disease.

The recommendations concerning imaging monitoring also depend on the therapeutic context. No pre-planned imaging monitoring was recommended in the case of mCSPC patients treated with ADT (as the only treatment or after docetaxel administration) or mCRPC patients treated with an ARTA or after having received docetaxel because such monitoring was considered necessary only if a patient experiences a clinical and/or biochemical relapse. The possibility of using regular twice yearly imaging monitoring in order to capture the best imaging response to treatment of ARTA-treated mCRPC patients was discussed, but the degree of consensus did not reach the threshold of acceptance. It was recommended that an imaging assessment should only be repeated at the end of docetaxel treatment in patients with mCSPC, and after 12 weeks’ treatment in patients with mCRPC. It is worth noting that, regardless of therapeutic context, it was always strongly recommended to use the same imaging techniques as those used at baseline, and all of the recommendations indicate that the preferred techniques are CT and BS. This preference reflects caution concerning new imaging techniques that are expected to be more sensitive than traditional techniques, but do not have standardized criteria for evaluating response. In any case, PET-PSMA and wbMRI are still only available at very few centers and are not widely used.

It is also worth noting that, in all but one clinical context, a number of factors were identified whose presence at baseline may indicate a different degree of disease aggressiveness, and should therefore be considered when planning monitoring frequency because some modifications to standard monitoring programs may be necessary. The only situation in which no such variables were identified was in the case of mCSPC patients who were candidates for docetaxel treatment, meaning that the presence of de novo metastases is per se a sign of aggressiveness and that no other factors need to be considered.

There are also some statements concerning the factors that may change an initially defined monitoring schedule: regardless of disease status (mCSPC or mCRPC) or the therapeutic context (chemotherapy, ADT or ARTA), these always included biochemical progression and the appearance or worsening of disease-related symptoms such as a worsening performance status, the occurrence of a skeletal event, and an increase in pain therapy.

Other aspects of mPC patient monitoring that are not strictly related to evaluating the course of the disease were also discussed, and there was strong agreement that testosteronemia should be evaluated whenever PSA levels increase, that standard monitoring plans should include regular assessments of bone health, and that metabolic factors should be regularly assessed, particularly in the case of patients at increased cardiovascular risk.

Clearly the clinical settings and treatment options addressed by the present Consensus are not able to fully cover all therapies that the quick evolution of PC management is progressively making disposable, requiring new editions of the Consensus. For example, monitoring procedures of treatment with abiraterone, which is approved for mCSPC in several European Countries but still not in Italy, or apalutamide and enzalutamide, which should have the same indication in the next future, will require specific discussions. Additionally, the use of apalutamide, darolutamide, and enzalutamide in a new disease setting, such as non metastatic CRPC, will open new challenges for the clinicians in defining the optimal monitoring procedures, which should be specifically addressed by new specific statements.

Considering the fields that were not addressed by the Consensus statements, the present paper is to be considered as having limitations. These limitations will be overcame by a new edition of the Consensus able to cover the fields, which the introduction of new active agents in PC landscape will open.

To this end, the results of the present Consensus Conference may be of help the clinicians in managing their PC patients. For example, the different approach followed in evaluating the monitoring of patients treated with docetaxel, and of those who receive an ARTA, could be valuable. From this point of view, for example, the St. Gallen Consensus considered the monitoring strategies regardless of the treatment strategy.

## 4. Materials and Methods 

As shown in Figure 1, the project was divided into four phases.

### 4.1. Definition of Questions Deserving Clarification/In-Depth Analysis

The Delphi technique [27] and the Nominal Group Technique (NGT) [28] are formal methods of reaching consensus that have been developed to overcome at least some of the negative aspects of group dynamics and ensure good decision making [29]. As they are considered equivalent in obtaining convergent opinions concerning a particular issue [30], this phase was carried out using the mini-Delphi technique [31] and involved a multidisciplinary Board of six experts (two medical oncologists, two radiotherapists, two urologists).

### 4.2. Review and Survey

On the basis of the questions deserving clarification/in-depth analysis identified in phase 1, the Board members:

Undertook a systematic review of the literature in the Medline, Embase and Cochrane databases, the ASCO, EAU, AIOM, and NCCN guidelines, and the St. Gallen Consensus Conference recommendations. 

Carried out an on-line survey of Urologists, and Medical and Radiation Oncologists, who were asked to choose from among various mPC management strategies.

### 4.3. Development of Statements

A specific statement was produced for each of the previously defined questions deserving clarification/in-depth analysis. Using a modified mini-Delphi approach, the statements were independently developed by each Board member, harmonized, and discussed during a final face-to-face meeting.

### 4.4. Consensus Conference

Selected clinicians belonging to the involved scientific societies took part in a Consensus Conference panel to which the Board members presented the final statements and explained the reasons for their choices on the basis of evidence or experience. All of the panelists then voted (Yes, No, Abstain) on each statement with the aim of reaching a consensus threshold of 80%; if this threshold was not reached, the statement was discussed, revised, and voted on again up to four times until it was.

The Consensus Conference panel included experts covering all of the specialties involved in treating mPC patients (Appendix A). The Consensus Conference was held in September 2018 and involved 69 experts, who routinely treated mPC patients and were all voting members.

## 5. Conclusions

Over the last ten years, the increasing availability of efficacious agents that can be sequentially used has made the management of mPC highly complex. In this scenario, clinicians have very little evidence on which to base the planning of an efficient monitoring program capable of capturing real disease progression in everyday clinical practice.

It is hoped that the recommendations made above, which were drawn up by experts following a formal and validated consensus procedure, will help clinicians face the everyday challenges of monitoring mPC patients.

## Figures and Tables

**Figure 1 cancers-11-01908-f001:**
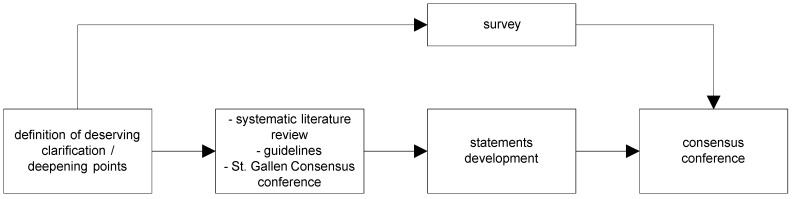
Project phases description.

**Table 1 cancers-11-01908-t001:** Summary of statements.

Statement	Timing	Factors
When should clinical and biochemical assessments be scheduled in the case of an mCSPC patient who is a candidate for ADT alone?	Every 12 weeks for the first 12 months, and every 24 weeks thereafter	
When should imaging assessments be scheduled in the case of an mCSPC patient who is a candidates for ADT alone?	In the case of a biochemical and/or clinical relapse (preferably CT and BS).	
Are there any factors that could influence the baseline monitoring plan of an mCSPC patient who is a candidate for ADT alone?		Age at the time of diagnosis, Gleason score, symptoms, the number and site(s) of metastases, the time of onset of metastases, time interval between radical local treatment and the onset of metastases.
Are there any factors that could change the initially defined monitoring schedule of an mCSPC patient being treated with ADT alone?		Trend of PSA levels and disease-related symptoms, (worsening in performance status, occurrence of a skeletal event, change in analgesic treatment).
When should clinical and biochemical assessments be scheduled in the case of of an mCSPC patient who is a candidate for treatment with ADT + docetaxel?	Every treatment cycle (clinical), at least at the third and sixth treatment cycle (biochemical).	
When should imaging assessments be scheduled in the case of an mCSPC patient who is a candidate for treatment with ADT + docetaxel?	At the end of docetaxel treatment using the same methods as those used at the time of the initial evaluation (preferably CT and BS).	
Are there any factors that could influence the baseline monitoring plan of an mCSPC patient who is a candidate for treatment with ADT + docetaxel?		No factor.
Are there any factors that could change the initially defined monitoring schedule of an mCSPC patient being treated with ADT + docetaxel?		Increasing PSA levels and worsening disease-related symptoms (worsening performance status, occurrence of a skeletal event, increased analgesic treatment).
When should clinical and biochemical assessments be scheduled in the case of an mCSPC patient without progressive disease who has concluded docetaxel treatment but is continuing ADT?	At least every 12 weeks.	
When should imaging assessments be scheduled in the case of an mCSPC patient who has concluded docetaxel treatment but is continuing ADT?	Only in the case of clinical and/or biochemical progression (preferably CT and BS).	
Are there any factors that could influence the baseline monitoring plan of an mCSPC patient who has concluded docetaxel treatment but is continuing ADT in the absence of progressive disease?		PSA level, appearance of symptoms, biological/clinical aggressiveness of the disease.
Are there any factors that could change the initially defined monitoring schedule of an mCSPC patient undergoing ADT who has been previously treated with docetaxel?		Increase in PSA levels and/or the onset or worsening of disease-related symptoms (worsening performance status, occurrence of a skeletal event, increase in pain therapy).
When should clinical assessments be scheduled in the case of an mCRPC patient who is a candidate for chemotherapy?	Every cycle.	
When should biochemical assessments be scheduled in the case of an mCRPC patient who is a candidate for chemotherapy?	At least every 6–8 weeks.	
When should imaging assessments be scheduled in the case of an mCRPC patient who is a candidate for chemotherapy?	After about 12 weeks using the same methods as those used for the baseline assessment (preferably CT and BS).	
Are there any factors that could change the initially defined monitoring schedule of an mCRPC patient during docetaxel treatment?		Increase in PSA levels and the onset or worsening of disease-related symptoms (worsening performance status, occurrence of a skeletal event, increase in pain therapy).
When should imaging assessments be scheduled In the case of an mCRPC patient who has completed chemotherapy and shows no signs of progression?	Depend on the results of clinical/biochemical assessments by using the same methods as those used for the baseline assessment (preferably CT and BS).	
When should clinical and biochemical assessments be scheduled in the case of an mCRPC patient who is a candidate for ARTA treatment?	PSA assessment every 12 weeks clinical evaluation every four weeks.	
When should imaging assessments be scheduled in the case of an mCRPC patient who is a candidate for treatment with an ARTA?	Should be based on the findings of clinical/biochemical assessments.	
Are there any factors that could influence the baseline monitoring plan of an mCRPC patient who is a candidate for ARTA treatment?		Site of metastases and disease-related symptoms.
Are there any factors that could change the initially defined monitoring schedule of an mCRPC patient undergoing ARTA treatment?		Trend of PSA levels and the onset of disease-related symptoms.
When should testosterone assessments other than the baseline assessment be scheduled in the case of patients with advanced prostate cancer (mCSPC/mCRPC)?	Testosterone evaluation every time there is an increase in PSA levels.	
When should bone health assessments other than the baseline assessment be scheduled in the case of patients with advanced prostate cancer (mCSPC/mCRPC)?	Standard monitoring plan should include regular bone health assessments.	
When should assessments of metabolic alterations other than the baseline assessment be scheduled in the case of patients with advanced prostate cancer (mCSPC/mCRPC) treated with ADT?	Regular metabolic assessments, particularly in presence of cardiovascular risk.	

Legend: ADT = androgen deprivation therapy – ARTA = androgen receptor targeting agent – BS = bone scan – CT = computerized tomography – mCRPC = metastatic castration resistant prostate cancer – mCSPC = metastatic castration sensitive prostate cancer.

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
