# Peer review of "Monitoring Patients with Metastatic Hormone-Sensitive and Metastatic Castration-Resistant Prostate Cancer: A Multidisciplinary Consensus Document"

_cancers, 2019, doi:10.3390/cancers11121908_

Round 1

Reviewer 1 Report

The authors and consensus conference PCa experts have undertaken a challenging clinical task to organize a comprehensive agreement on the frequency and methods used to monitor Italian men with mCSPC and mCRPC during the evolution of the disease for optimization of therapy.  The consensus team crafted and defined 12 statements each for mCSPC and mCRPC using available guidelines (AUA, EAU, ESMO, AIOM and NCCN), PCWG and published data including the clinical trials (STAMPEDE, LATITUDE, CHAARTED, PREVAIL), and St.Gallen Consensus Conference to make practical recommendations to monitor these men.  The therapies discussed for both diseases included ADT, chemotherapeutic agents cabazitaxel and docetaxel, enhanced androgen signaling inhibitors abiraterone acetate and enzalutamide, and bone microenvironment target radium 223.  The mCSPC statements were passed with a consensus of between 81% and 93%.  The mCRPC statements were passed with a consensus of between 84% and 94%.  Imaging was also discussed to evaluate patients. The preferred techniques were traditional CT and BS, however the new imaging techniques, PET-PMSA and wbMRI were discussed as more sensitive and incorporated in the future.  Overall this is a stimulating paper that will interest Oncologists in the USA and other European countries. However, I believe that established PCa oncologists may manipulate these recommendations to help personalize individual patient therapy.

Author Response

We would like to thank the reviewer for the constructive comments and criticisms.

We appreciate the comments and we certainly agree that PC oncologists may manipulate the consensus recommendations in tailoring the treatments. However, due to the lack of universally recognized guidelines the development of recommendations through a process based on formal procedures can be a good point of start in standardizing the monitoring procedures.

Reviewer 2 Report

Dr. Lapini and colleagues report a consensus statement on the monitoring of men with metastatic prostate cancer. Given the paucity of randomized data the topic if of great relevance in daily practice.

However, there are several issues that should be addressed.

Please indicate when the consensus conference took place, and how many of the 69 panellists routinely treat patients with metastatic prostate cancer patients themselves. Which of the listed participants of the consensus conference were voting members? The topic of the consensus document is metastatic hormone-sensitive (mCSPC) and metastatic castration-resistant prostate cancer (mCRPC). However, for mCSPC (sections 2.1 – 2.12) the authors only discuss question related to ADT +/- Docetaxel. Yet, they repeatedly cite the LATITUDE study but do not give any recommendations for patients treated with Abiraterone in this setting. Given that further options will become standard of care soon (Enzalutamid, Apalutamid) it would be of great interest to receive guidance for imaging of men receiving early endocrine treatment. Also, treatment options have been approved for non-metastatic CRPC (M0CRPC); in this setting recommendations with regards to diagnostic tests are warranted. I miss a critical appraisal of the literature including the role of role of PSMA-PET/CT and whole body MRI. Also, the authors are encouraged to discuss their consensus statements in the light of the APCCC 2017 consensus conference (Gillessen et al. Eur Urol 2018), which also addressed several of the questions/scenarios, discussed here. There is a lot of the redundancy in the text, which could be avoided by presenting a tabular overview on the recommended intervals of the respective diagnostic procedures in the various settings (of possibly some sort of flow-chart). Obvious recommendations, e.g. the repeatedly noted consensus that imaging should be done in case of symptoms suggesting disease progression could be added as a general note. This would really help the reader and make the consensus more practical. In 2.1. the authors recommend clinical +/- biochemical assessment of men receiving ADT for mCSPC every 12 weeks during the first year and then every 24 weeks. Given that median time to castration resistance is one year (STMAPEDE) it is hard to understand why intervals after 1 year - when the probability of disease progression is increasing - should be longer rather than shorter. Please explain. Please also explain “+/-“, i.e. when do you feel PSA should (not) be measured? In 2.10. the authors recommend no further imaging in patients with mCSPC who have completed chemotherapy with Docetaxel in the absence of clinical and/or biochemical progression. However, in this setting around 30% of progressions occurred without biochemical progression (Bryce AH et al.: ASCO 2018; Abstr. #5046). In 2.22. the authors write that testosterone should be evaluated every time there is an increase in PSA levels. What is the basis for this recommendation and which other guidelines recommend this? Many recommendations are rather vague, e.g. that (unspecified) PSA trends should be considered (2.21), regular bone health assessments (2.23) and metabolic assessments should be done. What is “regular” and what should these include (and what consequences should these assessments have)? It is somewhat unfortunate, that other questions arising in daily practice are not answered: Is it really necessary to order a bone scan and a CT scan when imaging is done? How might intervals be modified with advanced lines of treatment (resulting in lower response rates and shorter progression free survival).

Author Response

We would like to thank the reviewer for the constructive comments and criticisms. Unfortunately, since the consensus was based on a rigorous methodology it is impossible to change or to modify the described statements. Similarly it is impossible to add further statements.

1) in the new manuscript version the place of consensus was indicated; moreover we underlined that the all panelists treated mPC patients themselves and were voting members 

2) Since at the time of Consensus (and still today) the only approved treatment for mCSPC in Italy was docetaxel the statements concerned only this treatment. We agree that recommendations are of great utility also in the case of therapy with one new hormone agents (not only ABI but also APA and ENZ) which today are known as efficacious in mCSPC but these recommendations should be formally developed and discussed in a new consensus conference, with new statements defining the monitoring procedures for patients treated with these agents. The same for the treatments which will be shortly available for M0CRPC. These concepts were included in the new manuscript version.

3) The panelists expressed a high agreement in all statements concerning the type of imaging technique: their preference was for traditional techniques. This was summarized in the discussion where they were underlined the limitations of the new imaging techniques mainly due to the lack of standardised criteria for evaluating response and their still limited availability. On these bases we did not perform a critical appraisal of the literature.

4) The statements were discussed at the light of the APCCC 2017 consensus conference.

5) We agree. A tabular overview of the recommended imaging procedures was added to the manuscript.

6) From the conceptual point of view we agree that obvious repeated recommendations may be summarized as general note. However, from the methodological point of view the panelists recommendations cannot be modified and should be integrally reported. The tabular description of the statements could help the readers. 

7) We agree: the STAMPEDE results could suggest the adoption of a more frequent adoption of clinical and biochemical control to intercept the progression after the first year, nevertheless, the panelists’ statement reflected their experience and expertise. The panelists’ agreement was high and supported the 6-mo interval after the first treatment year. “+/-“ indicated that the PSA measurement is at discretion of the clinicians. (CONCORDATE?)

8) We agree that after CHAARTED a quote of progressions may occur without biochemical progression but they are usually characterized by symptoms appearance (e.g. clinical progression). Accordingly the panelists suggested an imaging assessment not only for PSA rising occurrence but also in the suspect of clinical progression.

9) the 2.22 recommendation is related to the panelists experience that in the case of biochemical progression, before of performing a treatment change, it should be mandatory to confirm the presence of androgen suppression.

10) the panelist preferred to indicated a PSA trend instead of one specific cut-off allowing the clinicians to tailor their strategy according to this parameter

11) the panelists did not indicate specific timing and / or technique for assessing bone health allowing the clinicians to perform this according to their practice. The main concept was that the bone health should be anyway assessed to adopt all strategies able to preserve it.

12) it is clear that many other statements are suggested by the daily clinical practice but it is impossible to address all of them in a single consensus. We thank you for suggesting these further arguments of debate.

Round 2

Reviewer 1 Report

Authors have addressed my concerns. 

Author Response

Thanks for your comments

Reviewer 2 Report

In the letter accompanying the revised version of the manuscript the authors state, with good reasons, that since the consensus was based on a rigorous methodology it seems impossible to change or to modify the described statements. For the same reasons, they feel that it is impossible to add further statements.

However, this leads to the situation that most of my concerns described in the initial review could not be addressed and thus remain. Whether the paper as it stands is of sufficient interest to an international readership or rather to the Italian medical community is to be judged by the editorial board of Cancers.

Author Response

Thanks for your comments. We reinforced the limitations of the manuscript in the discussion, underlying the need of a new consensus to address the points which were not evaluated by the statements of the current paper.